# Micro-scale fusion in dense relativistic nanowire array plasmas

Alden Curtis[1,3], Chase Calvi[2], James Tinsley[3], Reed Hollinger[1], Vural Kaymak[4], Alexander Pukhov[4], Shoujun Wang[1], Alex Rockwood[2], Yong Wang[1], Vyacheslav N. Shlyaptsev[1] & Jorge J. Rocca[1,2]

Nuclear fusion is regularly created in spherical plasma compressions driven by multi-kilojoule pulses from the world's largest lasers. Here we demonstrate a dense fusion environment created by irradiating arrays of deuterated nanostructures with joule-level pulses from a compact ultrafast laser. The irradiation of ordered deuterated polyethylene nanowires arrays with femtosecond pulses of relativistic intensity creates ultra-high energy density plasmas in which deuterons (D) are accelerated up to MeV energies, efficiently driving D–D fusion reactions and ultrafast neutron bursts. We measure up to $2 \times 10^6$ fusion neutrons per joule, an increase of about 500 times with respect to flat solid targets, a record yield for joule-level lasers. Moreover, in accordance with simulation predictions, we observe a rapid increase in neutron yield with laser pulse energy. The results will impact nuclear science and high energy density research and can lead to bright ultrafast quasi-monoenergetic neutron point sources for imaging and materials studies.

---

[1] Department of Electrical and Computer Engineering, Colorado State University, Fort Collins, CO 80523, USA. [2] Department of Physics, Colorado State University, Fort Collins, CO 80523, USA. [3] Nevada National Security Site, Las Vegas, NV 89030, USA. [4] Institut für Theoretische Physik, Heinrich-Heine-Universität Düsseldorf, 40225 Düsseldorf, Germany. Correspondence and requests for materials should be addressed to J.J.R. (email: jorge.rocca@colostate.edu)

Spherical compressions driven by multi-kilojoule lasers regularly produce fusion neutrons with typical yields on the order of $10^4 - 5 \times 10^5$ neutrons per joule[1–3]. Recent inertial confinement fusion (ICF) experiments at the National Ignition Facility used 1.9 megajoule laser pulses to produce a record $7.6 \times 10^{15}$ neutrons ($4 \times 10^9$ neutrons per joule) from deuterium–tritium fuel implosions[4]. In addition to ICF experiments, D–D fusion neutron bursts have been produced using energetic sub-ns pulses of a few hundred joules from chirped-pulse amplification lasers[5,6], and using petawatt class lasers[7–9]. However, all these experiments are limited to repetition rates of a few shots per hour or less. The ability to drive fusion reactions with compact lasers that can be fired at much higher repetition rates is of significant interest for fusion science, high energy density studies, and neutron pulse generation. Specific applications of neutron sources include neutron imaging and tomography[10], neutron scattering[11], and diffraction[12] for the study of material structure and dynamics, and neutron and neutrino detector development. An early experiment with a compact femtosecond laser demonstrated the generation of fusion reactions producing 140 neutrons per shot from a deuterated poly-ethylene flat target irradiated at an intensity of $10^{18}$ W cm$^{-2}$[13]. Since then, several different fusion target geometries, target densities, and laser irradiation conditions have been investigated using compact lasers. The targets used include deuterated thin films[6], cryogenic $D_2$[14], and deuterated clusters[9,15–20]. A significant advance in driving fusion reactions with compact lasers was the irradiation of deuterated clusters formed in gas jets with low energy femtosecond laser pulses, that allows for efficient volumetric heating of plasmas with an average ion density of $\sim 1 \times 10^{19}$ cm$^{-3}$[15] in which cluster explosions accelerates ions to multi-keV average energy[9]. Neutron generation efficiencies of $\sim 1 \times 10^5$ neutrons per joule were obtained in the form of short sub-ns bursts, a value similar to those obtained with multi-kilojoule laser. The ultrafast irradiation of ordered nanowire arrays share with nanoclusters the advantage of efficient volumetric heating, but have the additional advantage of creating a media with several orders of magnitude higher average plasma density[21]. We have recently shown that irradiation of aligned arrays of metallic nanowires with femtosecond laser pulses of relativistic intensity can volumetrically heat dense plasmas to multi-keV temperatures[21], reaching pressures only achieved in the laboratory in spherical compression with the world largest lasers[22].

In the following, we demonstrate that the interaction of laser pulses of relativistic intensity with aligned deuterated nanostructures accelerates ions up to MeV energies in near-solid-density media, opening a path to efficiently drive fusion reactions with joule-level lasers. We report a record in D–D neutron generation efficiency from plasmas generated by irradiating arrays of aligned high aspect ratio deuterated polyethylene (CD$_2$) with ultra-high contrast pulses of relativistic intensities from a compact laser.

## Results

Arrays of aligned high aspect ratio nanowires have vacant spaces surrounding the wires (Fig. 1a) that allow for the deep penetration of ultrafast optical laser pulse energy into near-solid-density material, where light is trapped and practically totally absorbed[21]. Electrons are ripped off the nanowire surface by the large laser field and are accelerated to high energy in the voids. These energetic electrons interact with the nanowires, rapidly heating the material to extreme temperatures, causing the nanowires to explode (Fig. 1b–d). Ions are rapidly accelerated between the nanowires, and the voids are filled with plasma, creating a

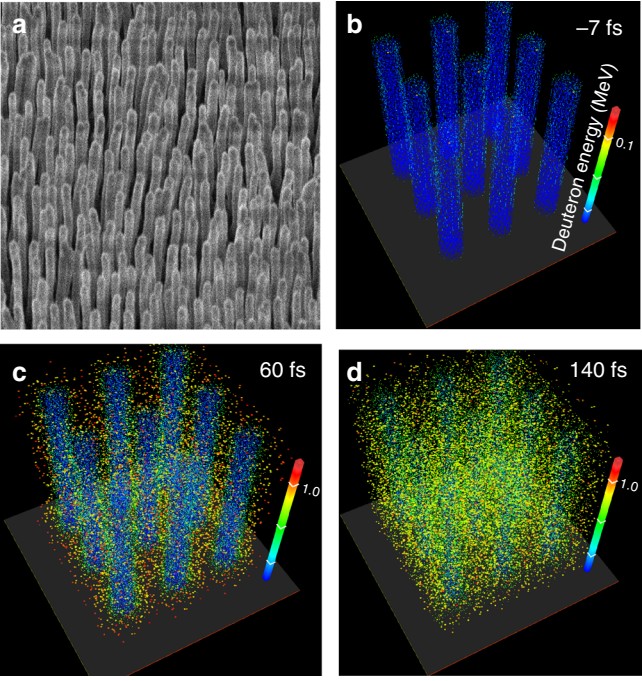

**Fig. 1** SEM image and 3D PIC simulation of energy distribution of deuterons. **a** SEM image of an array of 200 nm diameter CD$_2$ nanowires. **b–d** Three-dimensional particle-in-cell (PIC) simulation of the evolution of the energy distribution of deuterons in an array of 400 nm diameter CD$_2$ nanowires irradiated at an intensity of $8 \times 10^{19}$ W cm$^{-2}$ by an ultra-high contrast $\lambda = 400$ nm laser pulse of 60 fs FWHM duration. The laser pulses penetrate deep into the array where they rapidly heat the nanowires to extreme temperatures, causing the nanowires to explode (Fig. 1c, d). Deuterons are rapidly accelerated into the voids up to MeV energies, producing D–D fusion reactions and characteristic 2.45 MeV neutrons. Times are measured with respect to the peak of the laser pulse. The average density of the nanowire array corresponds to 16% solid density

continuous critical electron density layer that forbids further coupling of laser energy into the material (Fig. 1d). Assuming total laser energy absorption and volumetric heating of the target, the average energy per particle can be estimated to be:

$$E_{av} \approx \frac{a_0^2}{2} mc^2 \frac{n_c}{n_{av}} \frac{Z}{(Z+1)} \frac{c\tau}{L},$$

where $n_c = \varepsilon_0 m \omega_0^2 / e^2$ is the critical electron density (cm$^{-3}$), with the average particle density $n_{av}$, $\tau$ is the laser pulse duration, $Z$ is the mean ion charge, $L$ is the absorption depth in a target, and $m$ is the electron mass. The laser strength parameter, $a_0$, is defined as the normalized vector potential of the laser field and calculated as $a_0 = 0.855 \times 10^{-9} I^{1/2}$(W cm$^{-2}$) $\lambda_0$ (μm). For the conditions of the experiments discussed below ($a_0 \sim 3$, $\lambda_0 = 400$ nm, $n_{av} = 6.4 \times 10^{22}$ cm$^{-3}$, $Z = 2.67$, $\tau = 60$ fs, and $L \sim 5$ μm), $E_{av}$ can be calculated to be $\sim 0.6$ MeV. These energetic particles close the gaps between 400 nm diameter wires in an array with an average density corresponding to 15 percent solid density in <100 fs. After homogenization of the material, the plasma as a whole, with ions of mass $M$, begins to expand in the normal direction toward the laser pulse with a characteristic time scale $\tau_s \approx \frac{L}{c} \sqrt{\frac{Mc^2}{E_{av}}} \sim 1.5$ ps, but also toward the substrate, where the energetic deuterons moving into the target cause additional fusion reactions.

The use of sufficiently short laser pulses allows for very efficient coupling of the pulse energy deep into the nanowire array, heating to extreme temperatures a volume of near-solid-density material several microns in depth. This new approach to

volumetric plasma heating opens access to the ultra-high energy density plasma regime using compact joule-class femtosecond lasers that can fire repetitively. We show below that the irradiation of deuterated nanowire arrays with pulses of relativistic intensity can accelerate a large number of deuterons to energies near the peak of the D–D fusion cross-section, opening the possibility to efficiently drive D–D fusion reactions and generate bright quasi-monoenergetic ultrashort neutron pulses from a point source with compact high repetition rate lasers. The particle-in-cell (PIC) simulation results illustrated in Fig. 1 show the computed spatio-temporal energy distribution of energetic deuterons in an array of 400 nm diameter deuterated poly-ethylene ($CD_2$) nanowires irradiated at an intensity of $8 \times 10^{19}$ W cm$^{-2}$ by laser pulses of 60 fs duration. The average density of the target was assumed to correspond to 16% of solid density. The plasma is rapidly fully ionized and the electron density is computed to reach $6.4 \times 10^{22}$ cm$^{-3}$. The deuteron spectra calculated 60 fs after the peak of the laser pulse shows energetic ions with kinetic energy up to 3 MeV are generated. This energy greatly exceeds what we measured for D ions generated from solid targets irradiated under the same conditions. Moreover, the deep penetration of the heat in the nanowires results in a much larger volume of heated material, leading to the acceleration of a greater number of deuterons. In experiments conducted at these conditions, we observed a ~500 times increase in the number of 2.45 MeV D–D neutrons produced as compared to flat $CD_2$ targets irradiated with the same laser pulses. The highest yield shots produced $2.2 \times 10^6$ neutrons per joule of laser energy. This corresponds to more than an order of magnitude increase with respect to the favorable yield reported for deuterated clusters[9,15]. We have also observed a strong superlinear increase of the neutron yield as a function of the laser irradiation intensity. The PIC simulations predict a further increase in irradiation intensity to $5 \times 10^{20}$ W cm$^{-2}$ will shift the ion energy distribution to higher energies, practically depleting the population of the low energy deuterons and leading to a significant further increase in the neutron yield.

**Experiments and Simulations**. The experiments were conducted by irradiating arrays of aligned deuterated polyethylene ($CD_2$) nanowires with $\lambda = 400$ nm, ultra-high contrast ($>10^{12}$) pulses of 60 fs FHWM duration with energy up to 1.65 J. The experimental setup is schematically shown in Supplementary Figure 1. The laser pulses were generated by a frequency-doubled titanium: sapphire laser (Methods). The ultra-high contrast is necessary to prevent destruction of the nanowires prior to the arrival of the intense laser pulse. The laser pulses were focused at normal incidence onto the nanowire arrays using an f/1.7 parabolic mirror. The energy spectra of the plasma ions were recorded using a Thomson parabola spectrometer (TPS, see Methods) placed at 75 cm from the target along the target normal. Ions reached the TPS through a 100 µm hole on the axis of the focusing parabola. The primary diagnostics for neutron detection consisted of four EJ-228 plastic scintillator/photomultiplier detectors for time of flight measurements. In addition, two neutron bubble dosimeters were stationed outside the target chamber.

Measurements were conducted with arrays of aligned $CD_2$ nanowires either 200 or 400 nm diameter and ~5 µm in length. The average density of the arrays corresponded to 16 and 19% solid density, respectively. We developed a method to fabricate the arrays of aligned $CD_2$ nanowires by heated extrusion of $CD_2$ into porous membranes (Methods). Flat solid targets of the same $CD_2$ material and arrays of $CH_2$ were also fabricated and shot for comparison with every $CD_2$ nanowire array target. Fig. 2a shows a measured single-shot TPS spectrum from a flat $CD_2$ target

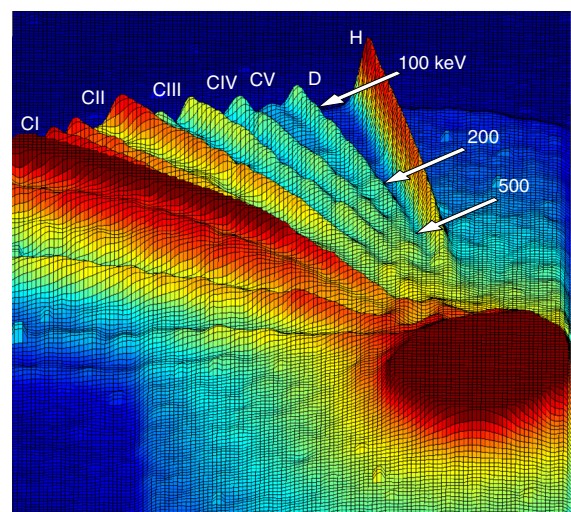

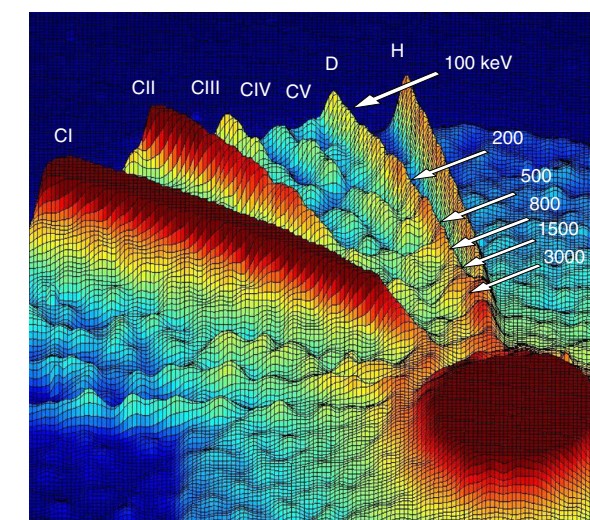

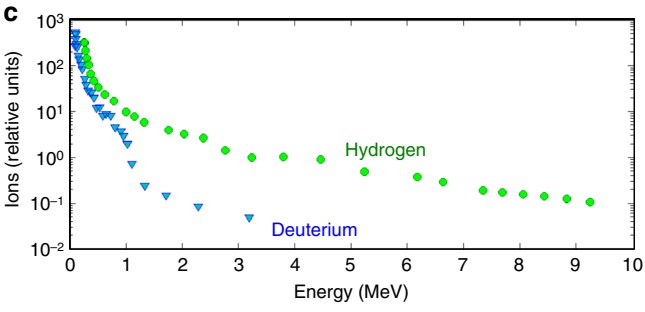

**Fig. 2** Measured single-shot Thomson parabola ion energy spectra: **a** flat solid $CD_2$ target irradiated at an intensity of ~$8 \times 10^{19}$ W cm$^{-2}$. Traces corresponding to H, D, and C ions are recorded. The tail of the deuteron energy spectrum reaches 0.5 MeV, **b** array of 400 nm diameter, 5 µm long, $CD_2$ nanowires. **c** Deuteron and proton energy distribution for the $CD_2$ nanowire array. The tail of the energy spectrum of the D and H spectra approaches 3 MeV and 10 MeV, respectively

irradiated at an intensity of $8 \times 10^{19}$ W cm$^{-2}$. Traces corresponding to D, H, and C ions are recorded. The energy of the deuterons in the flat solid target spectrum approaches 0.5 MeV. In comparison, arrays of $CD_2$ nanowires irradiated at the same conditions are observed to produce deuterons of up to 3 MeV energy and protons with energy as high as 10 MeV. A comparison of the ion energy spectra of $CD_2$ and $CH_2$ (Supplementary

Figure [2]) shows that the trace identified as corresponding to deuterons does not have a significant contribution of C VI ions, which have the same charge to mass ratio. The angular distribution of the fast deuterons was measured placing CR-39 plates at different angles from the target normal. To differentiate the deuteron flux from that of fast carbon ions, we covered part of each of the CR-39 plates with 2.5 or 4 μm thick Al foils that stop practically all carbon ions and let through deuterons with energies $\gtrsim$270 keV and $\gtrsim$400 keV, respectively. The deuteron flux was observed to peak in the direction of the target normal and to decrease as a function of angle, to become practically extinct for angles >67 degree (Supplementary Figure [3]). Figure 2c shows the deuteron and proton energy spectra extracted from Fig. 2b. The deuteron spectra is in good agreement with that resulting from the PIC simulations discussed below, and has a good overlap with the cross-section for D–D fusion reactions[23]. Consistently, time of flight traces recorded for several different target-detector distances show large neutron peaks at the expected arrival time for 2.45 MeV D–D neutrons (Fig. [3] and Fig. [4]). Data corresponding to solid $CD_2$ flat targets is also shown in Fig. [4].

When compared, the integrated neutron signals of the nanowire targets around 2.45 MeV are ~500 times larger. The maximum number of neutrons per shot was measured to be $3.6 \times 10^6$ for a laser pulse energy of 1.64 J, corresponding to $2.2 \times 10^6$ neutrons per joule, the largest fusion neutron yield obtained to date for joule-level laser pulse energies. If the same experiment with deuterated nanowires were to be conducted on a tritium containing substrate layer, the increase in the fusion cross-section combined with its shift towards lower ion energy, a significantly larger number of fusion neutrons could result.

Furthermore, the number of neutrons was measured to increase superlinearly with laser pulse energy (Fig. [5]). The rapid increase is in good agreement with the simulations we conducted using the deuteron ion energy distributions resulting from the PIC simulations and nuclear kinetics. For these measurements and simulations, the laser spot size was kept constant while the pulse energy was varied. The neutron yield is a function of the D–D fusion cross-section and the stopping power for deuterons in the target material. The simulations show the measured increase in neutron yield is caused by a shift of the deuteron energy distribution to higher energies. This computed deuteron energy increase is illustrated in Fig. [6] for arrays of $CD_2$ nanowires irradiated at intensities between $3 \times 10^{19}$ W cm$^{-2}$ and $1 \times 10^{21}$ W cm$^{-2}$. An increase of the intensity to $1 \times 10^{21}$ W cm$^{-2}$ is seen to nearly deplete the low energy deuterons shifting a significant fraction of the distribution to multiple MeV energy. The simulations also show that at the lowest irradiation intensities used in this experiment, ~$1 \times 10^{19}$ W cm$^{-2}$, the majority of the neutrons are produced within the plasma volume. In contrast, at the highest irradiation intensities explored, a significant fraction of the neutrons are produced in collision of accelerated deuterons with deuterium atoms in the material that surrounds the plasma. This includes neutrons generated by deuterons streaming into the ~200 μm thick $CD_2$ substrate layer. This fraction increases with the irradiation intensity. At intensities $>1 \times 10^{21}$ W cm$^{-2}$, the majority of the fusion reactions will occur in the substrate layer, outside the plasma volume. Therefore maximizing neutron

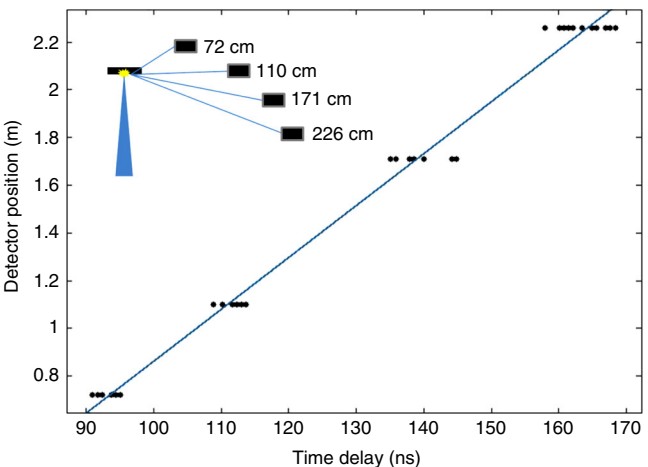

**Fig. 3** Time of flight neutron measurements. Time of flight PMT/scintillator neutron data from four detectors located at different distances from the target. The trigger was the plasma x-ray signal preceding the neutrons. The slope of the line fitting the data corresponds to the velocity associated with (2.48 ± 0.14) MeV neutrons

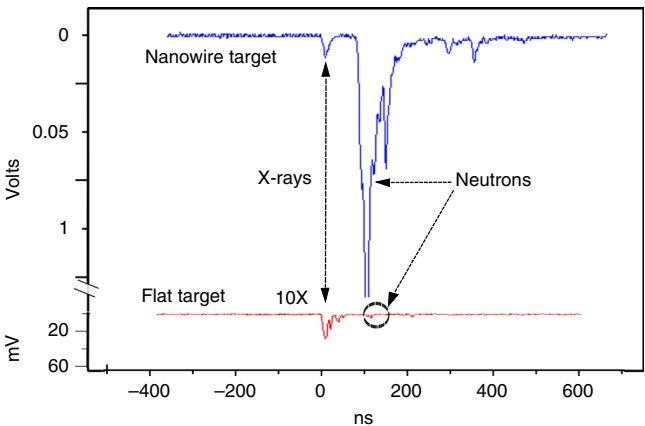

**Fig. 4** Comparison of time of flight neutron signals. Comparison of TOF neutron signals from a nanowire array target and a flat target. The flat target neutron signal (in red) was multiplied by 10 for clarity. The ratio of the average neutron yield of eleven nanowire shots to the average yield of six flat target shots at the same irradiation conditions yielded a value of 492

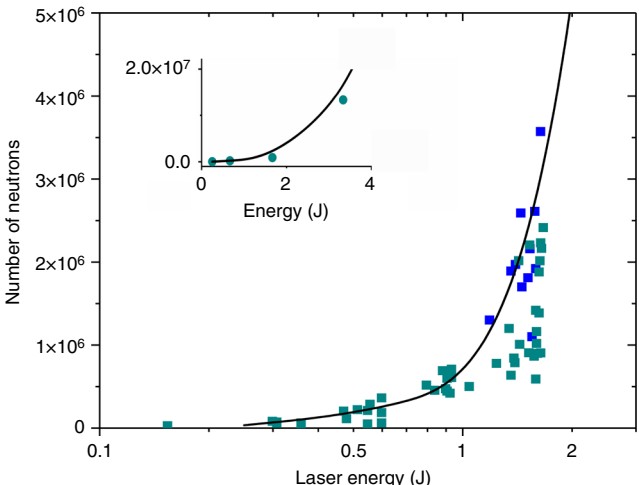

**Fig. 5** Neutron yield as a function of laser pulse energy on target. The dark blue squares are shots corresponding to a target with 200 nm diameter wires. All the other shots (light blue squares) are for targets consisting of 400 nm diameter wire arrays. Each point resulted from the average of four scintillator/PMT time of flight detector signals. The line shows the simulated energy dependence of the neutron yield calculated using deuteron energy distributions computed by the PIC model and nuclear kinetics. The inset extends the simulation to 3.5 J, where the green circles are computed values of the neutron yield

production at higher irradiation intensities will require the use of a thicker $CD_2$ substrate layer. Also, as the laser pulse energy is further increased beyond the values explored here, the optimum D–D neutron production might require a tradeoff between a further increase in the intensity and an increase in the irradiated volume. The higher intensities are also expected to generate a directed flux of high-energy deuterons that could be made to impinge in low Z convertors to drive "pitcher-catcher" neutron sources that have been demonstrated to create a large number of high-energy neutrons[6,24,25]. Finally, the simulations also show that the laser pulse drives a large forward electron current in the area around the wires. At higher irradiation intensities (eg. $5 \times 10^{21}$ W cm$^{-2}$), this forward current is computed to induce return current densities of tens of Mega-amperes per μm$^2$ through the nanowires[26]. The resulting strong quasi-static self-generated azimuthal magnetic field will pinch the deuterated nanowires into hot plasmas with a peak electron density exceeding 1000 times the critical density (Supplementary Figure 4).

In summary, we have realized a near-solid-density plasma regime in which deuterons from aligned nanostructures are accelerated up to MeV energies. The volumetric heating of aligned deuterated polyethylene nanowire arrays irradiated at relativistic intensity is shown to produce ultrashort neutron pulses with a ~500 times larger number of D–D neutrons than a deuterated flat solid target. A total of $2 \times 10^6$ neutrons per joule was generated, the largest D–D fusion neutron yield reported to date for plasmas generated by laser pulse energies in the 1 J range. A further increase of the irradiation intensity is predicted to shift the deuteron energy distribution to significantly higher energies, which can be expected to lead into a further increase in D–D fusion reactions. This volumetrically heated dense fusion environment that can be created at a high repetition rates with compact lasers is of interest for high energy density science and nuclear science. The approach can also lead to the efficient generation of ultrafast pulses of quasi-monoenergetic neutrons from a point source for time-resolved material studies, ultrafast neutron radiography, and spectroscopy, and for high-energy science applications such as neutrino detector development.

## Methods

**Experimental Setup**. Deuterated nanowire array targets were irradiated with ultra-high contrast femtosecond laser pulses of up to 1.65 J from a frequency-doubled, $\lambda = 400$ nm, chirped-pulse-amplification titanium:sapphire laser. The pulses had 60 fs FWHM duration. The ultra-high contrast necessary to prevent destruction of the nanowires prior to the arrival of the main pulse was achieved by frequency doubling the compressed pulses in a 0.8 mm thick type 1 KDP crystal. The conversion efficiency into second harmonic ($\lambda = 400$ nm) was ~40 percent, which means the generation of the highest energy ultra-high contrast second harmonic pulses (1.64 J) required fundamental wavelength pulses of ~6.5 J energy. The 400 nm wavelength second harmonic light was separated from the 800 nm fundamental beam using a sequence of four dichroic mirrors of 99.9 percent reflectivity at 400 nm (99.5 percent transmissive at the fundamental wavelength). The laser pulses were focused at normal incidence into a spot 2–2.6 μm diameter to achieve intensities up to $2 \times 10^{20}$ W cm$^{-2}$ on target using an f/1.7, 90 degree off-axis parabolic mirror. The focal spot size was determined by imaging with a ×50 objective onto a 12-bit CMOS camera. Approximately 30 percent of the energy was concentrated in the central spot; the laser pulse duration was measured using single-shot frequency resolved optical gating. The laser pre-pulse contrast in the picosecond range was monitored with a third-order scanning autocorrelator, and in the nanosecond range, it was measured using the combination of a silicon photodiode and a set of calibrated neutral density filters. The intensity contrast of the frequency-doubled pulse is inferred to be $>1 \times 10^{12}$. The laser pulse energy on target was monitored on a shot-by-shot basis by measuring a calibrated leak through a mirror with 99% reflectivity at $\lambda = 400$ nm, correcting for the reflectivity of the focusing parabola.

The ion energy distribution was measured using a TPS. The TPS built for these experiments uses two permanent Nd magnets, separated by a distance of 6 mm, creating a field of 5000 Gauss. The magnets are biased with a potential difference of up to 2400 V. A 6 mm diameter hole through the off-axis focusing parabola allows the ions to reach the TPS. The ions pass through a 100 μm pinhole placed at 50 cm from the plasma and through the collinear fields to impinge on a pair of matched microchannel plates (MCPs) stacked in chevron configuration. The spatial information in the emitted electrons is transferred into a phosphor screen deposited onto an optical fiber bundle, which couples the fluorescence out of the vacuum to be imaged onto a charge coupled device (CCD) (Andor CCD). The TPS

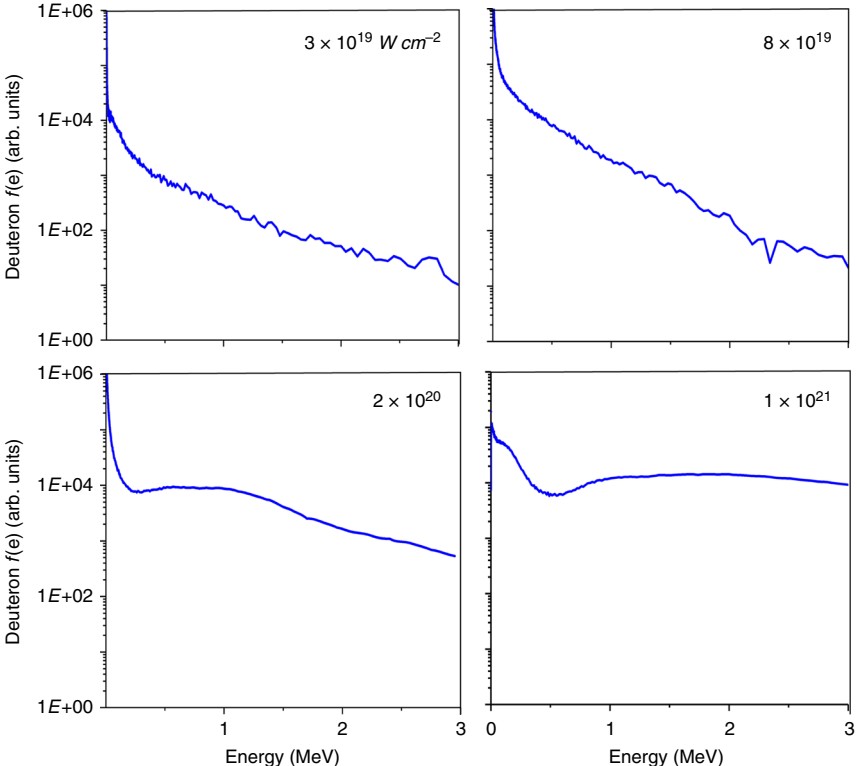

**Fig. 6** Deuteron energy spectra ($f(e)$, arb. units). Computed deuteron energy spectra for 400 nm diameter $CD_2$ nanowires at different irradiation intensities. The target average density corresponds to 16% of solid density

was calibrated for ion energy by gating the MCP with respect to the time of arrival of the laser pulse to the target. The primary diagnostics for neutron detection consisted of four time of flight neutron detectors comprising EJ-228 plastic scintillators coupled to photomultiplier tubes (Hamamatsu H2431–50). The detectors were shielded by 10 cm of Pb on the side facing the plasma, and by 5 cm of Pb on all other sides in order to reduce the X-ray signal reaching the photomultipliers. The neutron detectors were calibrated using the known neutron flux from a dense plasma focus at the National Security Technologies dense plasma focus facility[27]. The accuracy of the calibration is estimated to be ~25 percent, a value similar to the statistical fluctuations in the measurement of the number of neutrons. In addition, two neutron bubble dosimeters from Bubble Tech industries were stationed outside the chamber (with sensitivity 33 bubbles per mrem). The neutron number values inferred from the bubble counts corroborated the measurements from the scintillator detectors.

**Deuterated Nanowire Arrays**. The nanowire array targets were grown in house from deuterated polyethylene ($CD_2$) tested by infrared spectroscopy to contain >99% deuterium vs hydrogen. Arrays of 200 nm and 400 nm diameter wires ~5 μm in length were used in the experiments, with an average density corresponding to 16 and 19% solid density, respectively. The wires were formed by heated extrusion into ion-tracked polycarbonate or alumina porous membranes for the arrays of 400 and 200 nm diameter wires, respectively. The nanowire arrays were exposed by dissolving the membranes. A substrate that contains a ~200 μm thick $CD_2$ layer supports the nanowires. The morphology of each target array was characterized using scanning electron microscopy (SEM). The average target density can be determined by multiplying the density of the $CD_2$ material used by the fraction of volume occupied by the nanowires. The latter is known from the porosity of the templates used, and was confirmed by SEM images of the nanowire arrays. The nanowire targets have a diameter of 12.5 mm. To avoid shooting damaged regions, the target was displaced by 2 mm from the previous shot. This allowed us to acquire data from typically 12 shots per target. Flat $CD_2$ targets were fabricated by heating the same material under pressure in a hydraulic press.

**Simulation Tools**. The PIC simulations were conducted using the relativistic three-dimensional virtual laser-plasma laboratory (VLPL) code[28]. Its standard algorithms were extended by packages for optical-field ionization (OFI) and binary collisions, including electron impact ionization. OFI was treated as an under-barrier tunneling phenomenon in the static electric field[29,30] with only sequential field ionization considered. The probabilities for Coulomb collisions between all particles in one mesh cell were calculated by a binary collision package. PIC simulations utilized a three-dimensional geometry and self-consistently included ionization physics. A linearly polarized plane wave with 400 nm wavelength and Gaussian time envelope $a(t) = a_0 \exp(-t^2/\tau^2)$ was used to simulate the laser pulse where the normalized vector potential $a_0 = 3$ or 3.3 and a 60 fs FWHM pulse duration. The laser pulse was assumed to impinge on the deuterated polyethylene nanowire array at normal incidence. The PIC simulation space consisted of a cell volume encompassing the wires and inter-wire gaps, as well as space above the array to allow for expansion of the wire material as it explodes and thermalizes. The grid size used ranged from $50 \times 50 \times 1120$ to $100 \times 100 \times 2208$ on a mesh volume of $0.81 \times 0.81 \times 6.2$ μm$^3$. The time step was 0.00266 fs. The code accounts for local field enhancements, field fluctuations, and resonance heating. Simulations of neutron production dependence on laser irradiation energy were conducted with a post-processor code based on the code Radex[31] using the ion energy distributions computed by the PIC code. The neutron yield is a function of the D–D fusion cross-section and the stopping power of a deuteron in the target material; both of these are energy-dependent. The stopping power was used to calculate the distance traveled by a particle in the material as it decreases in energy from $E_i$ to $E_{i+1}$ due to multiple scattering. Using the cross-section corresponding to this energy range, we calculate the probability of fusion in that interval. Integration of these values from the initial energy gives the total fusion probability for that deuteron. The cross sections were taken from the Evaluated Nuclear Data File database[32]. The stopping power was calculated using SRIM[33]. In the simulation shown in Fig. 5, the laser spot diameter was an adjustable parameter assumed to be 5 μm. Accurate neutron generation modeling at intensities significantly beyond 2 J will require a new model that takes into account effects, which can be neglected at energies of the experiments reported here but that will play a role at significantly higher intensities, such as heating of the substrate material.

**Data Availability**. All relevant data are available from the authors.

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

## Acknowledgements

This material is based on work supported by the Air Force Office of Scientific Research under award number FA9560-14-10232 and FA9550-17-1-0278, and by Mission Support and Test Services, LLC. The simulations were conducted using the CU-CSU Sumit Computer Facility, NSF (ACI-1532235). We acknowledge the contributions of Zhenlin

Su and Jose Moreno in the development of the Thomson parabola and Conrad Buss for fabricating the nanowire targets.

## Author Contributions

J.J.R. and V.N.S. conceived the experiment. A.P. developed the PIC code. V.K. and A.P. preformed simulations and V.N.S. provided theoretical support. A.C., C.C., J.T., R.H., Y. W, S.W., and A.R. conducted the experiment, C.C. and A.C. fabricated the targets, J.T. calibrated the neutron detectors, Y.W., S.W., A.R., and J.J.R. developed the laser, and R. H., A.C., C.C., and J.J.R. developed the high contrast beamline. All the authors contributed to writing the paper.

## Additional information

**Competing Interests:** The authors declare no competing interests.

