## [Peer Review File · Nature Communications]

Reviewers' comments:

Reviewer #1 (Remarks to the Author):

The significant result of this work is twofold: first, the observation of an enhancement in D-D fusion neutron generation by short pulse irradiation of wire arrays compared to a flat target and, secondly, the apparently superlinear scaling of neutron yield with increasing pulse energy. The implications for the development of relatively small-scale intense neutron-generating facilities are clearly stated and could significantly enhance the research capabilities at a wide range of institutions. Beyond this there are implications for HED science in general and potentially for fusion energy research, if the scheme could be successfully scaled to a larger, higher-energy facility.

In my opinion the paper is well presented, internally consistent and highly relevant to several active areas of research, with a good level of detail provided that would enable another appropriately equipped group to attempt the same measurement. There are several technical points that I would like to see addressed before the paper is published, however.

The formulae presented are relevant and useful for interpretation of the results, although the authors do occasionally fail to define all of the variables they have used. It may be fair to assume that 'm' refers to the electron mass and that 'M' refers to the ion mass in the introduction, for example, but this should be made explicitly clear to avoid confusion. Given the wide readership, the normalized vector potential 'a' should be defined where first used for the benefit of non-specialists in the field. Likewise the units for particle density should be specified explicitly as cm^{-3} rather than left dimensionless.

Taking the authors' assumption that all of the laser energy is absorbed by the target (stated at ~ 1.8 J), one would assume that the initial total energy in the electrons would equal this value. If one takes the equation given by the authors for the average energy and multiplies it by the number of electrons in the volume one might expect to be heated (the laser spot size multiplied by the given heating depth of $5 \mu\text{m}$, multiplied by the stated average electron density), one arrives at a total energy of around 0.1 J. This suggests either an overestimate of the energy absorbed, an underestimate of the average energy, an underestimate of the electron density or an underestimate of the heated volume of the target. Would the authors be able to comment on this (admittedly very crude) calculation and suggest where the discrepancy may originate? Is the system simply too complex to characterize in this way? Was any characterization of the initial wire density possible, or was it assumed to be that of normal solid density CD2?

With regards to the experimental setup, the collecting solid angle of the Thomson parabola is very small. Did the authors take any data to examine how isotropic the ion spectrum was? Radiochromic film, CR-39 or image plate are frequently used for these wide-angle measurements – were any fielded? This would be important since the electric and magnetic fields near the target will be high, likely scattering low-energy ions more strongly than higher energy ones. There is also the potential for caustic 'filaments' in the ion distribution which may cause unreliable single-point measurements (see examples of cited in the introduction section of A. Macchi et al., Rev. Mod. Phys, Vol. 85, p.751 (2013)). Structures of this kind are likely to evolve unpredictably as the laser intensity increases.

The authors describe the superlinear scaling of total neutron yield with laser intensity (which is a key result); however, was the laser energy kept constant for these simulations, or was the focal spot kept the same and the laser energy increased? Figure 5 shows the relationship with pulse energy while Figure 6 and the main text refer to the changes brought about by increasing intensity – some additional clarification would be good here. It would be interesting to see how the neutrons per joule

metric the authors use in the introduction scales in each case, which could potentially be more informative as to the physics within the target. Is there an ideal trade-off between increasing energy while maintaining intensity (to illuminate a larger area of wires), or is it always better to have maximum intensity?

When the authors describe large return currents and pinching magnetic fields, are these seen in their simulations? How transient is the effect? If the laser drives a strong current into the target which is then neutralized by the return current, one may simplistically expect the net current density to be near-zero and thus no large persistent magnetic field would be generated. If the field is transient while the net current density is non-zero, does it persist long enough to drive hydrodynamics in the ions? What contribution to the field structure is made by the significant current-perpendicular density gradients present in the wire array before it explodes, and does the PIC simulation account for this? I appreciate that it is impossible to completely describe every aspect of the simulation, but as it stands this comment struck me as being less substantiated by evidence than the rest of the paper.

Do the authors believe that cold stopping powers from SRIM will be sufficiently accurate for the purposes of simulating ion transport in these highly-ionized dense plasmas? Has any assessment of the potential impact of stopping power modification been made?

With regard to the suitability of references, I found the paper supported its assertions well with relevant peer reviewed evidence and appropriately acknowledged the existing literature. I had only two suggestions:

A recent PRL published by G. Ren et al. ('Neutron Generation by Laser-Driven Spherically Convergent Plasma Fusion', PRL 118, 165001 (2017)) should probably be included when discussing high yields in terms of neutrons/joule of laser energy – while this occupies the upper end of the 10^4 - 10^5 n/J range the authors state in their introduction, it is a significant result that a reader may wish to compare to the present work.

Additionally, it is my understanding that very recent shots on NIF have exceeded 10^{16} neutrons of total yield – while I do not believe this result has yet been published, the authors may wish to check for an updated figure before submitting their final draft to ensure it is as current as possible. As it stands I believe the figure quoted by the authors is correct as the highest published yield.

Finally, there were a few typographical errors and minor clarifications needed for the figures:

- The SI unit 'joule' should not be capitalized when written in full
- Figure 3: what is the estimated timing uncertainty in the TOF measurements? What is the trigger source?
- Figure 4: no labelled y-axis; I presume this is a linear scale?
- Figure 4: The large peak on the blue trace is mis-timed by around 100 ns with respect to the largest peak in the red trace. Is this a genuine effect, or an artefact of the way the data has been plotted?
- Figure 5: It is difficult to distinguish between the squares and circles which makes it challenging to assess the true spread in the results. May I suggest using a different colour for one of them, to clearly separate the two data sets?
- Figure 5: the simulation results appear to agree more strongly with the smaller nanowires than the larger ones, even though the simulations the authors describe earlier in the paper were for 400 nm diameter wires. Do the authors have an explanation for this apparent behaviour? Were other simulations conducted where the difference in average density between the 200 nm and 400 nm wires was accounted for?
- In the methods section the target substrate is said to be 200 nm thick, while in the body of the report it is reported as 200 μ m thick. I presume one of them is a typo?

Overall, the general methodology of the experiment and analysis appear robust and well-designed. The presence of bubble detectors to corroborate the scintillator data is particularly welcome since short-pulse experiments can generate significant hard X-ray yields which can cause issues in scintillators. I find the conclusions that the authors draw from the data to be sound and believe that this paper will be of significant interest to a wide audience, especially among groups who have not previously had the capability to generate neutrons at these fluences. I look forward to seeing the finished article in print.

Reviewer #2 (Remarks to the Author):

Creation of a high-energy-density plasma by using a nanostructured target was applied to the production of an efficient fusion neutron source. The authors report significant increment of neutron yield by using nano-wire targets made of deuterated plastic (CD) compared to normal flat CD targets.

High intensity and high contrast laser pulse penetrates deeply into a CD target through the nano-CD-wire forest. Ultrafast volumetric heating of the CD nano-wires produces an overdense plasma. The high-intensity laser field is confined by the over-dense plasma and the bulk of the target. A large part of the laser energy is absorbed by the over-dense plasma and DD fusion reactions occur in the over-dense CD plasma and a CD target bulk. Physics related to the ultra-high-energy-density plasma creation with a nano-wire target is explained clearly based on the 2D-PIC simulation. Energetic (>3 MeV) deuteron generation is clearly observed both in the experiment and the simulation.

Fabrication of the nano-wire target seems to be easy, and this target can be supplied with a high repetition rate as rotating disc targets. The reviewer believes that this is a promising scheme to produce the laser-driven high-flux neutron source, however, the reviewer has several critical questions about the accuracy of the neutron diagnostics before the final decision.

1. This scheme is not applicable to the fusion power source but the neutron source. The reviewer requests the authors write several possible applications of the laser-driven neutron sources for broad readers of Nature Communications. This introduction helps the readers to understand importance and possibility of laser-driven neutron sources.
2. Figure 4 is very difficult to be understood. In the blue curve (wire target), there are several peaks at 330 ns, 450 ns, 510 ns, 680 ns, 740 ns. What do the peaks correspond? The reviewer guesses that the peak at 400 ns in the red curve (flat target) corresponds to the DD neutron pulse, however, the reviewer is confused by the fact that there is no peak at 400 ns in the blue curve. The reviewer can not judge whether their neutron diagnostics works properly or not only with the manuscript. The reviewer requests the authors define the time origin ($t = 0$ ns) in the Fig. 4.
3. Figure 3 also confuses the reviewer. Why do the neutron peaks move in the shots so much temporally?
4. There are no errors in the neutron yield evaluations. The authors used the four ToF detectors and two bubble detectors. I can not believe that all the detectors indicate the same neutron yields. There are several error sources, for example, an accuracy of calibration, statistical fluctuation of measured neutron number, the neutron signal difference between each detector.

Reviewer #3 (Remarks to the Author):

The manuscript presents very interesting results on neutron production using J-level lasers focused over nano-structured targets. The number of neutron produced in the proposed schema is very high for such a modest laser energy. If the scaling presented in figure 5 stands up to energies of few tens of J, this high repetition source could become a most interesting tool for scientific and industrial applications. The diagnostics included in the experiment (Thomson Parabola, Time of Flight neutron detector, bubble detectors) are carefully implemented and appropriated. I think that the results are grounded, innovative and relevant, and they deserve publication in Nature Communications. I have few remarks and suggestions which should be addressed before publishing the manuscript:

- 1) The energy of the 800 nm laser pulse is never stated. I think 1.64 J is the energy of the frequency-doubled pulse.
- 2) Which is the transverse size of the target? How many shots are you able to do in a single target?
- 3) Numerical parameters used in the simulation (mesh size, ppt, etc) should be listed in the methods section.
- 4) D spectra coming from the simulations (fig. 6) show a flattening of the energy distribution when raising the intensity. Do spectra recorded in the Thomson parabola show a similar trend?
- 5) The limit of validity of the scaling law of the neutron number on the laser energy (figure 5) is a most important question regarding the applications of this scheme. It will be very useful to continue the scan with simulations beyond 2 J.
- 6) Another remark about figure 5. The data coming from simulations is represented with a kind of fit. It will be better to include also the points corresponding to the real laser energy values simulated, and the neutron numbers found.

The very good quantitative agreement between experiments and simulations shown in this figure is puzzling. In laser-plasma interaction the agreement is usually poor, in contrast to here where it is excellent over a large range of laser energies (from 200 mJ to ~2 J). Are there some free parameters in the simulations?

Response to Referees

We are thankful to the Referees for their constructive comments. Below we reply to each one of their comments

Reviewer #1 (Remarks to the Author):

The significant result of this work is twofold: first, the observation of an enhancement in D-D fusion neutron generation by short pulse irradiation of wire arrays compared to a flat target and, secondly, the apparently superlinear scaling of neutron yield with increasing pulse energy. The implications for the development of relatively small-scale intense neutron-generating facilities are clearly stated and could significantly enhance the research capabilities at a wide range of institutions. Beyond this there are implications for HED science in general and potentially for fusion energy research, if the scheme could be successfully scaled to a larger, higher-energy facility.

In my opinion the paper is well presented, internally consistent and highly relevant to several active areas of research, with a good level of detail provided that would enable another appropriately equipped group to attempt the same measurement. There are several technical points that I would like to see addressed before the paper is published, however.

Reviewer #1: The formulae presented are relevant and useful for interpretation of the results, although the authors do occasionally fail to define all of the variables they have used. It may be fair to assume that ‘m’ refers to the electron mass and that ‘M’ refers to the ion mass in the introduction, for example, but this should be made explicitly clear to avoid confusion. Given the wide readership, the normalized vector potential ‘a’ should be defined where first used for the benefit of non-specialists in the field. Likewise the units for particle density should be specified explicitly as cm^{-3} rather than left dimensionless.

Reply: We have now explicitly defined in page 3 that “m” stands for the electron mass, and “M” for the ion mass. We have also added the definition of a_0 : “ a_0 is defined as the normalized vector potential of the laser field and calculated as $a_0 = 0.855 \times 10^{-9} I^{1/2} [\text{W}/\text{cm}^2] \lambda_0 [\mu\text{m}]$ ”. Likewise, we also added the missing units for the particle density.

Reviewer # 1: Taking the authors’ assumption that all of the laser energy is absorbed by the target (stated at ~ 1.8 J), one would assume that the initial total energy in the electrons would equal this value. If one takes the equation given by the authors for the average energy and multiplies it by the number of electrons in the volume one might expect to be heated (the laser spot size multiplied by the given heating depth of $5 \mu\text{m}$, multiplied by the stated average electron density), one arrives at a total energy of around 0.1 J. This suggests either an overestimate of the energy absorbed, an underestimate of the average energy, an underestimate of the electron density or an underestimate of the heated volume of the target. Would the authors be able to comment on this (admittedly very crude) calculation and suggest where the discrepancy may originate? Is the system simply too complex to characterize in this way? Was any characterization of the initial wire density possible, or was it assumed to be that of normal solid density CD2?

Reply: We thank the reviewer for having done a check on the magnitude of the energy per particle. Their comment made us realize we had not specified in the manuscript the fraction of the energy in the central spot. We now mention in the experimental set up section (page 8) that the laser spot contains ~ 30 percent of the laser pulse energy.

The apparent discrepancy mentioned by the reviewer arises from the combination of two factors: a) the fact that in these experiments the spot focus contained $\sim 30\%$ of the laser pulse energy; and b) the fact that the example given in the paper was , as stated in the manuscript, computed for the particular case of $a_0 = 3$, which corresponds to an intensity of $7.7 \times 10^{19} \text{ W cm}^{-2}$ and to a laser pulse energy of ~ 730 mJ (lower than the 1.64 J maximum energy used in the experiment).

$$E \sim 0.73 \text{ J} \times 0.3 = 0.23 \text{ J}$$

Taking into account that the resulting energy from formula [1] is 638 keV, (rounded to ~ 0.6 MeV in the manuscript) the kinetic energy for the sum of both electron and ions in a 2.5 micron diameter, 5 micron deep heated volume with an average charge of $Z= 2.67$ and an average electron density of $6.4 \times 10^{22} \text{ cm}^{-3}$ is:

$$E= 2.45 \times 10^{-11} \text{ cm}^3 \times 6.4 \times 10^{22} \text{ cm}^{-3} \times 6.38 \times 10^5 \text{ eV} \times 1.6 \times 10^{-19} \text{ J/eV} \times (Z+1)/Z = 0.22 \text{ J}$$

In summary, taking into account a) and b) above the apparent discrepancy is resolved.

In reference to the target density, the value used in the calculations is that of the nanowire array, and not the normal solid density of CD_2 . The following clarifying sentence was added in the Methods section (page 10): “The average target density can be determined by multiplying the density of the CD_2 material used by the fraction of volume occupied by the nanowires. The latter is known from the porosity of the templates used, and was confirmed by scanning electron microscope images of the nanowire arrays”.

Reviewer #1: With regards to the experimental setup, the collecting solid angle of the Thomson parabola is very small. Did the authors take any data to examine how isotropic the ion spectrum was? Radiochromic film, CR-39 or image plate are frequently used for these wide-angle measurements – were any fielded? This would be important since the electric and magnetic fields near the target will be high, likely scattering low-energy ions more strongly than higher energy ones. There is also the potential for caustic ‘filaments’ in the ion distribution which may cause unreliable single-point measurements (see examples of cited in the introduction section of A. Macchi et al., Rev. Mod. Phys., Vol. 85, p.751 (2013)). Structures of this kind are likely to evolve unpredictably as the laser intensity increases.

Reply: The experiments reported in the original manuscript did not field CR-39 plates. However we have welcomed the reviewer’s suggestion and have conducted a new experiment to determine how isotropic the ion distribution is. We placed CR-39 plates at different angles with respect to the target normal and exposed them to a single shot on target. The results are described in a paragraph added to page 6 of the manuscript, which reads:

“The angular distribution of the fast deuterons was measured placing CR-39 plates at different angles from the target normal. To differentiate the deuteron flux from that of fast carbon ions we covered part of each of the CR-39 plates with 2.5 μm or 4 μm thick Al foils that stop practically all carbon ions and let through deuterons with energies $\geq 270 \text{ keV}$ and $\geq 400 \text{ keV}$ respectively. The deuteron flux was observed to peak in the direction of the target normal and to decrease as a function of angle, to become practically extinct for angles > 67 degree (see Fig. S3 in the Supplemental Material). “

The following more detailed discussion was also added to the Supplemental Material:

“Fig. S3 shows the angular distribution of the fast accelerated deuterons measured positioning CR-39 plates (~ 0.5 cm x 1 cm size) at angles of 22.5 degrees, 45 degrees, 56 degrees and 67 degrees respect to the target normal. The detectors were placed at different distances to avoid saturation (overlap of holes), and the deuteron flux values were computed correcting for the respective geometric factors. Fractions of the CR-39 plates were covered with 2.5 μm or 4 μm thick Al foils to differentiate D from C ions. These foils are transparent to D ions with energy above ~ 270 KeV and ~ 400 KeV respectively, but stop all carbon ions with kinetic energy $< 2.5 \text{ MeV}$ and $< 3.9 \text{ MeV}$ respectively. By placing a 2.5 μm Al foil in front of the Thomson parabola we corroborated that all carbon ions are stopped from reaching the CR-39 detector plates. Consequently, only deuterons and H impurity ions are recorded. Fig. S1(a) shows optical microscope photographs of CR-39 plates with 4 μm thick Al filters positioned at three different angles after they were exposed to a single laser shot and developed for two hours. The laser pulse energy on target was 1.38 J. The angular distribution of the ion flux (number of ions per unit area reaching the detectors) resulting from this measurement is illustrated in Fig S1(d). While the focusing parabola did not allow us to place a CR-39 detector plate in the direction normal to the target, it can be seen that the maximum deuteron flux occurs at the smallest angle respect to the target normal. The flux decreases as a function of angle to become nearly extinct for angles > 67 degree. The angular distribution of deuterons measured with both Al foil filters was practically the same.”

Fig. S3. Images of exposed CR-39 plates placed at different angles with respect to the target normal (a-c). (d) D flux as a function of angle. The nanowires were 200 nm diameter and the energy of target was 1.38 J.

Reviewer #1: The authors describe the superlinear scaling of total neutron yield with laser intensity (which is a key result); however, was the laser energy kept constant for these simulations, or was the focal spot kept the same and the laser energy increased? Figure 5 shows the relationship with pulse energy while Figure 6 and the main text refer to the changes brought about by increasing intensity – some additional clarification would be good here. It would be interesting to see how the neutrons per joule metric the authors use in the introduction scales in each case, which could potentially be more informative as to the physics within the target. Is there an ideal trade-off between increasing energy while maintaining intensity (to illuminate a larger area of wires), or is it always better to have maximum intensity?

Reply: We added a sentence in page 6 clarifying that as the laser pulse energy was increased the laser spot was maintained constant. In the range of energies investigated the neutron generation efficiency was observed to increase strongly with increased laser intensity, as the kinetic energy and range of the deuterons increase. However, as the pulse energy is further increased in future studies the trade-off mentioned by the reviewer between increased intensity and increased irradiated volume can be expected take place. To point this put the following sentence was added in page 7 of the manuscript:

“Also, as the laser pulse energy is further increased beyond the values explored here, the optimum D-D neutron production might require a tradeoff between a further increase in the intensity and an increase in the irradiated volume.”

Reviewer #1: When the authors describe large return currents and pinching magnetic fields, are these seen in their simulations? How transient is the effect? If the laser drives a strong current into the target which is then neutralized by the return current, one may simplistically expect the net current density to be near-zero and thus no large persistent magnetic field would be generated. If the field is transient while the net current density is non-zero, does it persist long enough to drive hydrodynamics in the ions? What contribution to the field structure is made by the significant current-perpendicular density gradients present in the wire array before it

explodes, and does the PIC simulation account for this? I appreciate that it is impossible to completely describe every aspect of the simulation, but as it stands this comment struck me as being less substantiated by evidence than the rest of the paper.

Reply: We have conducted new PIC simulations to illustrate the generation and spatio-temporal evolution of extremely large quasi-static magnetic field, and the pinching of the nanowires for the specific case of CD_2 nanowires. Current-perpendicular density gradients and all other effects are self-consistently taken into account in the simulations. The results are summarized below, and in what is now a new figure and text added to the supplemental section.

Fig. S4. PIC simulation showing the spatial distribution of the magnetic field (a-d) and electron density (e-h), for four different times ($t = -44$ fs, $t = -10.67$ fs, $t = 1.33$ fs, $t = 14.67$ fs respectively) with respect to the laser pulse. The 400 nm diameter nanowires are irradiated with an intensity of 5×10^{21} W/cm² ($\lambda = 400$ nm). The return current is observed to generate a strong quasi-static magnetic field that pinches the nanowires. The electron density is in units of critical density, $n_c = 7 \times 10^{21}$ cm⁻³.

‘The PIC simulations shows that the current produced by the forward acceleration of the electrons in the inter-wire gaps by the laser field the via $\vec{v} \times \vec{B}$ force induces a large return through the nanowires. At significantly higher irradiation intensities (eg. $5 \times 10^{21} \text{ W cm}^{-2}$) than those used in the experiments reported here, this return current is computed to reach current densities of tens of Mega-amperes per μm^2 . This return current will in turn result in the generation of a strong quasi-static self-generated azimuthal magnetic field which evolution is illustrated in Fig. S4 (a-d) for the case of CD_2 nanowires irradiated by a laser pulse of 30 fs duration with an intensity of $5 \times 10^{21} \text{ W cm}^{-2}$. The magnitude of this magnetic field is several gigaGauss. The return current lasts for the duration of the laser pulse. The resulting Lorentz force alters the plasma hydrodynamics pinching the deuterated nanowires into hot plasmas with a peak electron density exceeding 1000 times the critical density (Fig. S4 e-h) ‘.

Reviewer #1: Do the authors believe that cold stopping powers from SRIM will be sufficiently accurate for the purposes of simulating ion transport in these highly-ionized dense plasmas? Has any assessment of the potential impact of stopping power modification been made?

Reply: The volume of the plasma is quite small, of the order of $20 \mu\text{m}^3$. In spite of its very high density, at the highest laser pulse irradiation energies explored that are of most interest, only a small fraction of the D-D fusions take place in the plasma itself. Specifically, the range of 200 KeV and 1 MeV deuterons in CD_2 is 2.4 μm and 16 μm respectively, as compared with the laser spot radius of $< 1.3 \mu\text{m}$. Consequently, the majority of the fusions occur when an energetic deuteron from the plasma interacts with a deuteron at rest in the surrounding material. This is the case that is modeled using input from SRIM.

Reviewer #1: With regard to the suitability of references, I found the paper supported its assertions well with relevant peer reviewed evidence and appropriately acknowledged the existing literature. I had only two suggestions:

A recent PRL published by G. Ren et al. (‘Neutron Generation by Laser-Driven Spherically Convergent Plasma Fusion’, PRL 118, 165001 (2017)) should probably be included when discussing high yields in terms of neutrons/joule of laser energy – while this occupies the upper end of the 10^4 - 10^5 n/J range the authors state in their introduction, it is a significant result that a reader may wish to compare to the present work. Additionally, it is my understanding that very recent shots on NIF have exceeded 10^{16} neutrons of total yield – while I do not believe this result has yet been published, the authors may wish to check for an updated figure before submitting their final draft to ensure it is as current as possible. As it stands I believe the figure quoted by the authors is correct as the highest published yield.

Reply: The paper of G. Ren et al. was added to the manuscript as Ref. 3. We have not found a paper reporting higher yield in the very recent NIF shots, but would be happy to also add it if we become aware of it.

Reviewer #1 points

reply : Finally, there were a few typographical errors and minor clarifications needed for the figures:

- The SI unit ‘joule’ should not be capitalized when written in full . **Reply:** the correction was made.
- Figure 3: what is the estimated timing uncertainty in the TOF measurements? What is the trigger source?
Reply: We modified the caption of Fig. 3 to note that the trigger was the x-ray /gamma ray pulse . The uncertainty on the time of arrival is negligible respect to the neutron time of flight.
- Figure 4: no labelled y-axis; I presume this is a linear scale?

Reply: the y-axis label was added to Fig.4, which has a linear scale.

- Figure 4: The large peak on the blue trace is miss-timed by around 100 ns with respect to the largest peak in the

red trace. Is this a genuine effect, or an artifact of the way the data has been plotted?

Reply: The largest peak in the blue trace corresponding to the nanowire target is due to neutrons, while the largest peak in the red trace corresponding to the flat target is the x-ray peak. To avoid this confusion we have now labeled the peaks in Fig. 4. Please notice that the x-ray peaks occur at the same time in both cases.

- Figure 5: It is difficult to distinguish between the squares and circles which makes it challenging to assess the true spread in the results. May I suggest using a different colour for one of them, to clearly separate the two data sets?

Reply We modified Fig. 5 such that now the squares and circles also have different color tonality.

- Figure 5: the simulation results appear to agree more strongly with the smaller nanowires than the larger ones, even though the simulations the authors describe earlier in the paper were for 400 nm diameter wires. Do the authors have an explanation for this apparent behaviour?. Were other simulations conducted where the difference in average density between the 200 nm and 400 nm wires was accounted for?

Reply: The computations were conducted assuming the same average density.

- In the methods section the target substrate is said to be 200 nm thick, while in the body of the report it is reported as 200 μm thick. I presume one of them is a typo?

Reply: The target substrate thickness is 200 μm . We corrected the typo in the methods section.

Overall, the general methodology of the experiment and analysis appear robust and well-designed. The presence of bubble detectors to corroborate the scintillator data is particularly welcome since short-pulse experiments can generate significant hard X-ray yields which can cause issues in scintillators. I find the conclusions that the authors draw from the data to be sound and believe that this paper will be of significant interest to a wide audience, especially among groups who have not previously had the capability to generate neutrons at these fluences. I look forward to seeing the finished article in print.

Reviewer #2 (Remarks to the Author):

Creation of a high-energy-density plasma by using a nanostructured target was applied to the production of an efficient fusion neutron source. The authors report significant increment of neutron yield by using nano-wire targets made of deuterated plastic (CD) compared to normal flat CD targets.

High intensity and high contrast laser pulse penetrates deeply into a CD target through the nano-CD-wire forest. Ultrafast volumetric heating of the CD nano-wires produces an overdense plasma. The high-intensity laser field is confined by the over-dense plasma and the bulk of the target. A large part of the laser energy is absorbed by the over-dense plasma and DD fusion reactions occur in the over-dense CD plasma and a CD target bulk. Physics related to the ultra-high-energy-density plasma creation with a nano-wire target is explained clearly based on the 2D-PIC simulation. Energetic (>3 MeV) deuteron generation is clearly observed both in the experiment and the simulation.

Fabrication of the nano-wire target seems to be easy, and this target can be supplied with a high repetition rate as rotating disc targets. The reviewer believes that this is a promising scheme to produce the laser-driven high-flux neutron source, however, the reviewer has several critical questions about the accuracy of the neutron diagnostics before the final decision.

Reviewer #2: 1. This scheme is not applicable to the fusion power source but the neutron source. The reviewer requests the authors write several possible applications of the laser-driven neutron sources for broad readers of Nature Communications. This introduction helps the readers to understand importance and possibility of laser-

driven neutron sources.

Reply: We modified the introduction of the manuscript to list several possible neutron source applications. A sentence was added in page 2, that reads: “Specific applications of neutron sources include neutron imaging and tomography¹⁰, neutron scattering¹¹ and diffraction¹² for the study of material structure and dynamics, and neutron and neutrino detector development.” Please see manuscript for references.

Reviewer #2: 2. Figure 4 is very difficult to be understood. In the blue curve (wire target), there are several peaks at 330 ns, 450 ns, 510 ns, 680 ns, 740 ns. What do the peaks correspond? The reviewer guesses that the peak at 400 ns in the red curve (flat target) corresponds to the DD neutron pulse, however, the reviewer is confused by the fact that there is no peak at 400 ns in the blue curve. The reviewer can not judge whether their neutron diagnostics works properly or not only with the manuscript. The reviewer requests the authors define the time origin ($t = 0$ ns) in the Fig. 4.

Reply: We have redrawn fig 4 such that the origin of time ($t=0$) now corresponds to the arrival of the laser pulse to the target. We have also added clarifying labels to Fig. 4. A label now indicates that to the peak at 330 ns corresponds to x-rays/gamma rays. The peak occurring at the same time in the red trace corresponding to the flat target is now also identified as x-rays/gamma rays. The main peak at 400 + ns in the nanowire target trace (blue curve) is the main neutron pulse and is now identified as such. The neutron signal for the flat target is very small and is now also identified. The latter much smaller peaks are likely caused by neutrons that are scattered and are re-directed towards the detector after losing energy via elastic or inelastic collision.

Reviewer #2: 3. Figure 3 also confuses the reviewer. Why do the neutron peaks move in the shots so much temporally?

Reply: The neutrons generated have an energy spread that results from momentum transfer from the fast deuterons, and from scattering. However, the measured average neutron velocity corresponds to a neutron energy of (2.48 ± 0.14) MeV, that agrees very well the 2.45 MeV from D-D fusion reactions.

Reviewer #2: 4. There are no errors in the neutron yield evaluations. The authors used the four ToF detectors and two bubble detectors. I cannot believe that all the detectors indicate the same neutron yields. There are several error sources, for example, an accuracy of calibration, statistical fluctuation of measured neutron number, the neutron signal difference between each detector.

Reply: We added a sentence to the caption of Fig.5 indicating that each of the neutron numbers plotted are an average of the readings of the four scintillator detectors. We also added a sentence in the Methods section (p. 9) specifying that the uncertainties calibration of the neutron detector performed with a dense plasma focus is estimated to have an uncertainty 25 percent, which along with the statistical fluctuations of similar value constitute the dominant error.

Reviewer #3 (Remarks to the Author):

The manuscript presents very interesting results on neutron production using J-level lasers focused over nano-structured targets. The number of neutron produced in the proposed schema is very high for such a modest laser energy. If the scaling presented in figure 5 stands up to energies of few tens of J, this high repetition source could become a most interesting tool for scientific and industrial applications. The diagnostics included in the experiment (Thomson Parabola, Time of Flight neutron detector, bubble detectors) are carefully implemented and appropriated. I think that the results are grounded, innovative and relevant, and they deserve publication in Nature Communications. I have few remarks and suggestions which should be addressed before publishing the manuscript:

Reviewer #3 1) The energy of the 800 nm laser pulse is never stated. I think 1.64 J is the energy of the frequency-doubled pulse.

Reply: A sentence was added in the Experimental set-up section (page 8) to specify the conversion efficiency

into second harmonic ($\lambda=400$ nm) is $\sim 40\%$, which implies that the generation of the highest energy ultra-high contrast second harmonic pulses (1.64 J) required fundamental wavelength pulses was ~ 6.5 J energy.

Reviewer #3 2) Which is the transverse size of the target? How many shots are you able to do in a single target?

Reply: We added the following sentence in page 10 of the Methods section: “The nanowire targets have a diameter of 12.5 μm . To avoid shooting over damaged regions of the target, the target was displaced by 2mm between shots. This allowed us to acquire data from typically 12 shots per target”

Reviewer #3 3) Numerical parameters used in the simulation (mesh size, ppt, etc) should be listed in the methods section.

Reply: We added the following sentence in page 11: ‘The grid size used ranged from 50 x 50 x 1120 to 100 x 100 x 2208 on a mesh of 0.81 x 0.81 x 6.2 μm^3 with 0.00266 fs step size’

Reviewer #3 4) D spectra coming from the simulations (fig. 6) show a flattening of the energy distribution when raising the intensity. Do spectra recorded in the Thomson parabola show a similar trend?

Reply: The Thomson parabola data was taken for irradiation intensities below the values at which the flattening is predicted to occur. The experimental study of the deuteron energy distribution at higher intensities is part of future work.

Reviewer #3 5) The limit of validity of the scaling law of the neutron number on the laser energy (figure 5) is a most important question regarding the applications of this scheme. It will be very useful to continue the scan with simulations beyond 2 J.

Reply: We added an inset in Fig. 5 showing the predicted neutron generation for energies up to 3.5 J. However, it should be noticed that accurate modeling of neutron generation at intensities significantly beyond 2 J will require a new model that takes into account effects, such as heating of the substrate material, which can be neglected at energies below 2 J but that will play a role at significantly higher intensities. In addition, optimum neutron generation at higher intensities will also require altering the target design to take full advantage of the increased deuteron range (eg. use of a thicker CD_2 substrate). A sentence was added in page 11.

Reviewer #3 6) Another remark about figure 5. The data coming from simulations is represented with a kind of fit. It will be better to include also the points corresponding to the real laser energy values simulated, and the neutron numbers found.

Reply: The points from which simulation curve was obtained were added to the inset in Fig.5.

Reviewer #3. The very good quantitative agreement between experiments and simulations shown in this figure is puzzling. In laser-plasma interaction the agreement is usually poor, in contrast to here where it is excellent over a large range of laser energies (from 200 mJ to ~ 2 J). Are there some free parameters in the simulations ?

Reply: The laser spot diameter was the only adjustable parameter. To acknowledge this we have added the following sentence in page 11 : “In the simulation of Fig.5 the laser spot diameter was an adjustable parameter assumed to be 5 μm ”.

REVIEWERS' COMMENTS:

Reviewer #1 (Remarks to the Author):

I am satisfied that the authors have gone to significant lengths to address the points raised in the initial review. The revised manuscript covers these well, providing ample evidence in support of the new content. The revised figures are now much clearer.

It was especially pleasing to see that the authors went to the trouble of collecting additional experimental data with the addition of CR-39 plates, which itself may prove interesting as a test of the TNSA-enhancing potential of these targets. In my opinion the paper is now ready to publish and should be of great interest to the Nature Communications readership.

Reviewer #2 (Remarks to the Author):

The authors revised properly the manuscript according to the suggestions and comments given by the reviewers. I suggest to the editor this revised manuscript to be published in Nature Communications.

Reviewer #3 (Remarks to the Author):

This reviewer provided confidential remarks to the editor recommending publication.

Response to referees

“Micro-Scale Fusion in Dense Relativistic Nanowire Array Plasmas”

by A. Curtis et al

The referees comments are included below. The referees have not brought any new comments. Therefore a reply is not required.

REVIEWERS' COMMENTS:

Reviewer #1 (Remarks to the Author):

I am satisfied that the authors have gone to significant lengths to address the points raised in the initial review. The revised manuscript covers these well, providing ample evidence in support of the new content. The revised figures are now much clearer.

It was especially pleasing to see that the authors went to the trouble of collecting additional experimental data with the addition of CR-39 plates, which itself may prove interesting as a test of the TNSA-enhancing potential of these targets. In my opinion the paper is now ready to publish and should be of great interest to the Nature Communications readership.

Reviewer #2 (Remarks to the Author):

The authors revised properly the manuscript according to the suggestions and comments given by the reviewers. I suggest to the editor this revised manuscript to be published in Nature Communications.

Reviewer #3 (Remarks to the Author):

This reviewer provided confidential remarks to the editor recommending publication.